# Anti-Ballistic Performance of PPTA/UHMWPE Laminates

**DOI:** 10.3390/polym15102281

**Published:** 2023-05-12

**Authors:** Long Zhu, Weixiao Gao, Dmitriy A. Dikin, Simona Percec, Fei Ren

**Affiliations:** 1Department of Mechanical Engineering, Temple University, Philadelphia, PA 19122, USA; long.zhu@temple.edu (L.Z.); weixiao.gao@temple.edu (W.G.); ddikin@temple.edu (D.A.D.); 2Temple Materials Institute, Temple University, Philadelphia, PA 19122, USA; simona.percec@temple.edu

**Keywords:** PPTA, UHMWPE, body armor, interlayer adhesion, failure mechanism

## Abstract

Poly(*p*-phenylene terephthalamide) (PPTA) and ultra-high-molecular-weight polyethylene (UHMWPE) are high-performance polymer materials largely used for body armor applications. Although composite structures from a combination of PPTA and UHMWPE have been created and described in the literature, the manufacture of layered composites from PPTA fabrics and UHMWPE films with UHMWPE film as an adhesive layer has not been reported. Such a new design can provide the obvious advantage of simple manufacturing technology. In this study, for the first time, we prepared PPTA fabrics/UHMWPE films laminate panels using plasma treatment and hot-pressing and examined their ballistic performance. Ballistic testing results indicated that samples with moderate interlayer adhesion between PPTA and UHMWPE layers exhibited enhanced performance. A further increase in interlayer adhesion showed a reverse effect. This finding implies that optimization of interface adhesion is essential to achieve maximum impact energy absorption through the delamination process. In addition, it was found that the stacking sequence of the PPTA and UHMWPE layers affected ballistic performance. Samples with PPTA as the outermost layer performed better than those with UHMWPE as the outermost layer. Furthermore, microscopy of the tested laminate samples showed that PPTA fibers exhibited shear cutting failure on the entrance side and tensile failure on the exit side of the panel. UHMWPE films exhibited brittle failure and thermal damage at high compression strain rate on the entrance side and tensile fracture on the exit side. For the first time, findings from this study reported in-field bullet testing results of PPTA/UHMWPE composite panels, which can provide important insights for designing, fabricating, and failure analysis of such composite structures for body armors.

## 1. Introduction

Lightweight protective body armors have been highly desired by soldiers and security personnel to enhance their performance [1]. The basic requirement of body armor is to protect against high-speed projectiles and strikes with sharp objects. Other aspects, such as lightweight, comfort, and wearability are also important considerations to the wearers. Hard body armors made of ceramics or metal plates may provide better protection from high-speed bullets, but they are usually heavier and rigid [1]. In contrast, soft body armors made from high-performance polymer fibers are flexible, lightweight, and comfortable, while providing a certain level of protection in everyday use outside the combat zone [1,2]. Although significant progress has been made in improving the performance of soft body armor by adopting high-performance fabrics, there is still a great challenge in developing body armor with low bulge deformation, antiballistic capabilities, and flexibility at the same time [3,4]. In recent years, nanomaterials, such as carbon nanotubes (CNTs) and graphene, which are among the stiffest and strongest materials, have been explored as reinforcements for armor composites [5,6]. New designs and material systems are highly desired for further improvement [7].

With the rapid development in modern technology, soft body armors are expected to have advanced functions in addition to resisting bullets, such as response to various mechanical stimuli, monitoring health condition, and service reliability in harsh environments [8,9]. Shear-thickening gels (STGs) have gained extensive attention as a new kind of intelligent material for improving the anti-impact performance of soft body armor, thanks to their unique impact-buffering capacity and low modulus [10]. By incorporating STGs into composites, multifunctional soft body armors have been fabricated that can sense mechanical forces, maintain body temperature, monitor human physical state, and perform other functions [10,11,12]. To cite a few examples, Fan et al. [10] prepared a SiO_2_/shear-thickening gel/reduced graphene oxide (RGO)@Kevlar fabric composite with excellent strain sensing ability, fire resistance, electro-heating, and anti-impact performance. Zhao et al. [11] prepared a conductive shear thickening gel-Kevlar fabrics (c-STG/Kevlar) body armor material that exhibited both mechano-sensing and anti-impact capabilities. Another work by Fan et al. [12] involved the fabrication of a nano-SiO_2_/carbon nanotube (CNT)/shear-thickening gel@polyurethane composite, which exhibited resistance to hazardous liquids (strong acid and alkali), excellent stain-sensing ability, and anti-impact capabilities.

Literature studies show that fiber-reinforced composite materials have broad applications in soft body armors [13,14,15,16,17,18]. The reinforcement provided by high-strength fibers, including glass fibers, carbon fibers, poly(*p*-phenylene terephthalamide) (PPTA) fibers, ultra-high-molecular-weight polyethylene (UHMWPE) fibers to various matrices, such as epoxy, polypropylene, polyurethane, phenolics, and low-density polyethylene (LDPE) can help achieve excellent mechanical properties, such as high strength-to-weight ratio, good energy absorption capability and fracture toughness, and superior flexibility.

One particularly important parameter in the design of armor-grade reinforced materials is the interfacial adhesion of the fiber to the matrix [19]. The interfacial adhesion and interphase properties govern the load transfer between the composite components [20], and the delamination process, which plays a significant role in impact energy absorption [21,22,23]. Different strategies have been implemented to tune the interlayer adhesion between composite components for ballistic applications. For example, Kessler et al. [19] conducted surface treatment on E-glass fibers using different dispersion approaches to adjust the fiber-matrix bonding of the glass/epoxy laminate. Naveen et al. [24] introduced graphene nanoplatelets to the Kevlar^®^/Cocos nucifera sheath-reinforced composites to enhance the fiber and epoxy matrix adhesion. Wang et al. [25] utilized adhesive polyurethane to enhance the adhesion between yarns in aramid/polyurethane composite for ballistic applications.

While various studies have demonstrated that certain levels of increase in interlayer adhesion are favorable for absorbing higher levels of impact energy [19,24,25,26,27,28], other studies suggested that strong interlayer adhesion might reduce the impact energy absorption. For example, Wang et al. [25] illustrated that during the projectile penetration process, strong adhesion could cause the yarn breakage without being pulled out, which was detrimental to impact energy absorption. Zhang et al. [26] conducted finite element modeling (FEM) to investigate the effect of interface strength on the performance of cross-ply UHMWPE laminate plates containing Dyneema^®^ SK76 fiber and polyurethane matrix. The results showed that with the increase of the interface normal strength from 1.2 MPa to a high level of 1200 MPa, the ballistic performance of the laminates deteriorated dramatically, evidenced by the significant increase in the residual projectile velocity. The authors contended that the laminates with strong interface strength are subjected to high bending stress, leading to premature failure of the rear surface of the laminates [26].

As previously mentioned, among various high-performance polymers, PPTA (e.g., Kevlar^®^) [29,30,31], and UHMWPE, such as Spectra^®^ or Dyneema^®^ [32] have been extensively utilized for ballistic applications due to their remarkable mechanical properties. While many studies were dedicated to composite materials consisting of either PPTA or UHMWPE, a few of them focused on the ballistic performance effect by combining the two. Hofsté et al. [33] fabricated composites from chopped PPTA fibers and UHMWPE powder. Mechanical properties characterization showed that there was a large difference between the experimentally obtained values and the theoretically predicted values. They attributed this difference to the voids in the composite and weak adhesion between the PPTA fiber and the UHMWPE matrix. The same authors used chromic acid to oxidize the UHMWPE powder before mixing it with the PPTA fiber [34]. The results showed that the mechanical properties of the composites were improved because of stronger bonding between PPTA fiber and UHMWPE matrix. However, degradation problems arose due to the instability of the oxidized UHMWPE. Li et al. [35] used silane, which contains both hydrophobic and hydrophilic groups, to surface modify the PPTA fibers in PPTA/UHMWPE powder composites. The results demonstrated that the silane modification process enhanced the tensile strength and wear resistance of the composites. More recently, Guleria et al. [36] developed a microwave-assisted compression molding process to prepare PPTA/UHMWPE power composites. Mechanical properties characterization showed that PPTA/UHMWPE composites exhibited higher ultimate tensile strength, flexural and hardness properties, and impact energy absorption rate, compared to pure UHMWPE.

Knitting PPTA and UHMWPE yarns was another way of fabricating PPTA/UHMWPE composites in addition to preparing PPTA fiber/UHMWPE powder-based materials [37,38]. Adhesive resins such as ethylene vinyl acetate [39], vinyl ester [40], and epoxy [41] as well as other components such as carbon nanotubes [41] were used to enhance the bonding between the PPTA fabrics and UHMWPE fabrics/fibers. However, this type of fabrication process does not form direct bonding between the PPTA and the UHMWPE. Moreover, complex processing steps, such as mixture/solution preparation, sample impregnation, heating, curing, and drying, were required in these processes. To the best of the authors’ knowledge, continuous UHMWPE films were not used for fabricating PPTA/UHMWPE composites. The UHMWPE films in our fabrication act as a bonding layer between the PPTA fabrics, which eliminated the use of additional adhesives or components. Thus, the use of UHMWPE films can greatly simplify the process of manufacturing layered structures compared to using UHMWPE powders and other technologies.

As previously mentioned, both PPTA and UHMWPE are important high-performance polymers for lightweight body armor. By combining these two materials together, it is expected that the synergy between PPTA and UHMWPE can be achieved, and that the antiballistic performance can be further enhanced beyond what either material can achieve on its own. In this study, laminate panels consisting of PPTA fabrics and UHMWPE films were produced using a previously developed simple method, which included plasma treatment and hot-pressing [42]. The plasma treatment process greatly enhanced the interlaminar adhesion between PPTA and UHMWPE layers. This improvement is expected to initiate enough delamination during bullet penetration, which can consume impact energy. This is superior to the case where weak or no interlaminar adhesion exists between the layers. The ballistic performance of the laminate samples was investigated by bullet testing. The results were correlated with the interlaminar adhesion reported in our previous work [42]. Failure mechanisms of the PPTA and UHMWPE layers were studied with the assistance of electron microscopy.

## 2. Materials and Methods

### 2.1. Materials

PPTA fabrics woven from Kevlar^®^ 49 fiber (area density: 218 g/m^2^, thickness: 0.37 mm, ends × picks/10 cm: 67 × 67, weave: plain, part number: AR30-FC-000160, Goodfellow Corporation, Coraopolis, PA, USA) and UHMWPE films (thickness: 0.2 mm, part number: ET30-FM-000100, Goodfellow Corporation, Coraopolis, PA, USA) were acquired from a commercial source. The properties provided by the vendor are shown in Appendix A.

### 2.2. Sample Preparation

Prior to the plasma treatment, UHMWPE films (6” × 6”) were first cleaned with acetone, then rinsed with deionized water, and dried in an oven at 50 °C overnight. Oxygen plasma treatment of UHMWPE films and PPTA fabrics (6” × 6”) at different durations was performed using a Harrick Plasma Cleaner (PDC-32G, power: 18 W). In our previous work [42], we conducted characterizations of UHMWPE and PPTA after plasma treatment. The SEM and AFM results revealed that the plasma treatment process had cleaned and roughened the surfaces of UHMWPE and PPTA, which enhanced the mechanical interlocking between the two. Additionally, the FTIR and XPS analyses demonstrated that the plasma treatment process had introduced active functional groups, which could help the bonding between the two [42]. These surface modifications resulted in an improved interlaminar adhesion between UHMWPE and PPTA, as shown by the 180-degree peel test results. Furthermore, by adjusting the plasma treatment time, controllable interlamellar adhesion between PPTA fabrics and UHMWPE films was achieved [42].

After the plasma treatment process, the UHMWPE films and PPTA fabrics were stacked alternatively, as shown in Figure 1. The hot-pressing was conducted at 190 °C for 1 h using a heat press machine (Across International, Livingston, NJ, USA). The interlayer compression was adjusted using a mechanical knob on the top of the hot-pressing machine, which was calibrated in advance using a thin film force sensor (FlexiForce sensor, Tekscan Inc., Boston, MA, USA). For all samples, the pressure was controlled to be about 173 kPa during the hot-pressing process.

Two different stacking sequences were used (Figure 1): (ⅰ) eight layers of UHMWPE films and nine layers of PPTA fabrics (PE(8)-KF(9)), in this case, PPTA fabric was on the outer surface, and (ⅱ) nine layers of UHMWPE films and eight layers of PPTA fabrics (PE(9)-KF(8)), in this case, UHMWPE film was on the outer surface. Two samples were prepared for each condition. The mass of all samples is around 80 g.

### 2.3. Testing and Characterization

Ballistic testing was performed on UHMWPE/PPTA laminates according to NIJ standards [43,44]. The bullet used in this study was 0.22 lr, which has an average velocity of 1255 feet per second as determined by the chronograph (Appendix A). The shooting distance was fixed at 4.572 m (15 ft). Each sample was attached to a 6” × 6” wooden box containing Plastalina modeling clay (Craft Smart^®^, Michaels Stores, Inc., Irving, TX, USA) (Figure 2 and Appendix A). Six shots were conducted on each sample. After the ballistic testing, the geometry of the bullet-induced indentations in the clay (Figure 2d) was measured to evaluate the energy absorption capability of the laminate samples, including their diameters and depths. The maximum and minimum indentation diameters were recorded and averaged. In Figure 2d, the check marks denote partial penetration shots, whereas the cross marks indicate full penetration shots. As shown in Figure 2d, shots #3 and 6 partially penetrated, showing indents with large diameters and shallow depths, whereas indents of completely penetrated shots #1, 2, 4, and 5 showed smaller diameters and greater depths.

Back face signature (BFS), which is usually measured from the backing clay, is a particularly important indicator for evaluating the performance of body armors [29]. If BFS exceeds a certain limit, the wearer could be harmed even if the bullet was stopped by the body armor. According to NIJ standards [43,44], complete penetration or indentation depth of BSF in backing clay greater than 44 mm should be classified as a failure. In this study, the failure rate was calculated by dividing the number of shots with indentation depth greater than 44 mm by the total number of shots. The indentation volume is another vital indicator to assess energy absorption capability, but it is usually challenging to measure [45].

For failure analysis, the laminate samples were sectioned by a sharp knife around the penetration holes. The cross sections, the bullet entrance surfaces, and the bullet exit surfaces near the penetration holes were examined using scanning electron microscopy (SEM, Quanta 450 FEG, Thermo Fisher-FEI, Hillsboro, OR, USA).

## 3. Results and Discussion

### 3.1. Anti-Ballistic Performance

#### 3.1.1. Effect of Interlayer Adhesion

The results of the ballistic testing are summarized in Figure 3. The different degrees of interlayer adhesion obtained as a result of the treatment of UHMWPE and PPTA with oxygen plasma at different exposure times were described in our previous study [42]. As shown in Figure 3, samples prepared by simply laying plasma untreated PPTA fabrics ((u)KF) and without hot pressing showed poor ballistic performance with a failure rate of 0.33 and an average penetration depth of 38.67 mm. Adding UHMWPE films between the layers of PPTA fabric ((u)PE-(u)KF*) improved the ballistic performance with a failure rate of 0.23 and an average penetration depth of 28.60 mm. In contrast, laminate samples made by hot-pressing PPTA fabrics and UHMWPE films without plasma treatment ((u)PE-(u)KF) showed a further improvement in ballistic performance with a failure rate of 0.17 and an average penetration depth of 26.10 mm. During hot pressing, the UHMWPE films were melted, which led to a certain level of adhesion between the layers of PPTA fabrics [42]. This favorably affects the ballistic performance of composites compared to samples without interlayer adhesion ((u)KF and (u)PE-(u)KF*) (Figure 3d). These results demonstrated that a certain level of interlayer adhesion increases the absorption of ballistic impact energy. When UHMWPE was plasma-treated for 1 min, the interlayer adhesion improved, and the samples exhibited the best ballistic performance with the lowest failure rate of 0.05 and an average indentation depth of 21.71 mm in the backing clay. However, samples containing plasma-treated PPTA fabrics and showing higher interlaminar adhesion did not exhibit better ballistic capability, as evidenced by a higher failure rate and a greater average indentation depth.

Samples (u)KF and (u)PE-(u)KF*, which were prepared by simply stacking PPTA and/or UHMWPE layers together without hot-pressing, exhibited no interlaminar adhesion. For all other samples, bonding was achieved by hot pressing. In these cases, delamination was expected to occur to absorb additional impact energy. In our previous work [42], we found that by introducing an oxygen plasma treatment process, both the topology and chemistry of both PPTA and UHMWPE surfaces were altered. Consequently, composites of PPTA fabrics and UHMWPE films with improved and controllable interlaminar adhesion were prepared (as shown in Figure 3d). By comparing the ballistic performance with the interlaminar adhesion, it can be seen that moderate interlaminar adhesion improves the ballistic performance of the laminated panels, which is consistent with literature data [19,24,25,26,27,28], as discussed in the previous section. Moderate interlayer adhesion can help to absorb impact energy by enabling delamination and debonding [28]. However, if the interlayer adhesion is too weak, such as observed in the untreated samples (u)PE-(u)KF, then the delamination process is insufficient to consume a large amount of impact energy of the high-speed projectile. Conversely, strong adhesion will cause the projectile to perforate directly with very limited delamination or even without delamination, which does not contribute to energy absorption either.

#### 3.1.2. Effect of Stacking Sequence

To investigate the effect of stacking sequence of PPTA and UHMWPE, two types of samples were made as shown in Figure 1: (i) samples with eight layers of UHMWPE and nine layers of PPTA or PE(8)-KF(9) samples, and (ii) samples with nine layers of UHMWPE and eight layers of PPTA, or PE(9)-KF(8) samples. Figure 4a–c show the failure rate, average indentation depth, and diameter of three groups of samples with the two stacking sequences. The PE(8)-KF(9) samples exhibit better ballistic performance, with a lower failure rate. This may be attributed to the following reasons. First, the first layers of the PE(8)-KF(9) samples are PPTA fabrics, which could provide better resistance against high-speed bullets due to its stronger mechanical properties than UHMWPE (Appendix A). Second, the PPTA yarns could entangle the bullet and effectively reduce its spin. Lastly, PPTA has a much higher upper working temperature (thermal stability) than UHMWPE, which avoids the thermal damage (often noticed in UHMWPE) and may also contribute to the better performance of the PE(8)-KF(9) samples.

Many factors affecting the impact performance of soft body armors are discussed in the literature. Mawkhlieng et al. [2] classified various parameters into the following categories, including (a) material parameters, such as fiber modulus, tenacity, density, and yarn to yarn friction [46]; (b) structural parameters, such as the number of layers, yarn twist, thread density, weave; (c) projectile parameters, such as mass, shape and velocity, and (d) testing parameters, such as shooting location, angle boundary conditions, and the number of shots, etc. In addition, the layering sequence was also considered to be an important factor affecting the performance of laminates. Park et al. [47] investigated the role of layering sequences in the penetration resistance of unidirectional (UD)/woven fabric hybrid panels. The cross-plied UD fabrics used in their study are UHMWPE fiber-based Dyneema^®^ SB31 and *p*-aramid fiber-based Gold Flex^®^. The woven fabric is a plain weave of 600 denier *p*-aramid yarns [47]. The authors claimed that when the unidirectional (UD) and woven fabric of hybrid panels were sequenced in the order of decreasing stiffness, the perforation resistance against infrangible bullets (5.56 mm NATO fragment-simulating projectile (FSP)) was improved, which was attributed to the less restraint of the subsequent rear-component layers. However, sequencing the component layers in a reverse manner enhanced the blunt trauma resistance against frangible bullets (0.44 caliber (10.9 mm) magnum semi-jacketed hollow point (SJHP)). They believed that this resulted from the better coupling of yarn elongation in the frontal and rear component layers [47]. Subramaniam et al. [48] studied the effect of stacking configuration on the quasi-static penetration performance of kenaf/glass hybrid fiber metal laminates and concluded that laminates with glass fiber plies on the outer surface exhibited better penetration resistance. They claimed that when the high strength and high stiffness glass fiber plies were placed on the outer layer, the energy absorption rate increased due to the higher strength and elongation required to induce the fracture of the flexible glass plies. Thus, the contact stress from the indenter was effectively transferred before it propagated to subsequent layers. However, when kenaf fabric was placed as an outer layer, premature failure occurred before the stress was transferred to the woven structure due to the low shear force required for penetration [48]. O’Masta et al. [32] contended that when the ply with the highest compressive strength was placed at the entrance side, and the ply with the highest tensile strength was placed at the back side of the laminate, the multi-material laminates exhibited significant impact performance advantages given the mechanics of progressive projectile penetration accompanied by the transformation of out-of-plane compression into in-plane tension [32].

In this study, two factors—the interlaminar adhesion and stacking sequence—were examined in terms of their effect on the ballistic performance. The results showed that a moderate increase in interlayer adhesion between PPTA and UHMWPE layers can improve the ballistic performance of the laminates. Laminates with PPTA as the outmost layer performed better than those with UHMWPE as the outmost layer. Future work will include the investigation of more parameters, including other stacking structures.

### 3.2. Failure Mode Analysis

#### 3.2.1. Failure Analysis of Fully Penetrated Samples

Failure of PPTA fabrics in PE(9)-KF(8) samples

Understanding the projectile penetration mechanism is of critical importance for designing high-performance body armors. The laminate sample (u)PE(9)-(u)KF(8) was sectioned around a completely penetrated hole (Figure 5a) and imaged by SEM. Figure 5b shows the layered structure of the panel with alternating layers of UHMWPE and PPTA, and the micro delamination (marked with white arrows) located outside the penetration hole. The fiber bundles were bent along the direction of the projectile’s motion (indicated by yellow and blue arrows). Figure 5c displays the obtuse tips of the fractured fibers, which were often found on the entrance side of the target (yellow arrow), indicating the rupture mechanism as a result of shear stresses (inset of Figure 5c). Shear stress concentrated on the sides of the projectile during penetration and shear cutting failure is usually the first failure mode in ballistic impact. For very high-speed impact, shear failure could be the only failure mechanism, where the damage is concentrated around the penetration hole [49]. At locations close to the exit side of the laminate (Figure 5d,e), the fibers split (defibrillate), pulling out their constituents and elongating, as illustrated in the inset of Figure 5d, indicating predominant rupture due to tensile stress caused by the large deflection of the bulge [50]. Deceleration of the projectile during penetration is the main reason for the change in the failure mode [49]. After the velocity falls below the critical speed of the shear failure, subsequent layers may peel off and bend, and fibers can break under the tensile forces. Results of mechanical defibrillation and fiber splitting can be also observed on the exit side (Figure 5f).

Figure 6 shows the morphology of PPTA fibers in another tested laminate sample (u)PE(9)-(5)KF(8). The damaged PPTA fibers on the entrance side (Figure 6a) exhibit kinks (Figure 6b) possibly due to pinching by neighboring fibers or buckling induced by compressive stress resulting from the stretch-and-release of raptured fiber (Figure 6c). Failure of the sample starts instantaneously due to high contact stresses applied by the projectile on the entrance side. Figure 6d depicts a large bulge and enlargement of the penetration hole (in comparison to the entrance side (Figure 6a) on the exit side of the sample, which is also observed in UHMWPE composite laminates [23] and signifies the different failure mechanism between the entrance and exit sides. When the projectile progressed towards the rear face of the composite panel with decreasing velocity, the load could be distributed into a wider area, resulting in localized bending and bulging around the impact zone. It is thus believed that materials with a high out-of-plane Young’s modulus can reduce the bending deformation and delay the occurrence of tensile failure on the exit side of the panel. Figure 6e,f present fiber stripping and necking characteristics, respectively, indicating that the fibers experienced high tensile stress before failure. This can also be observed in the cross-section around the penetration hole of the sample (u)PE(9)-(u)KF(8) (Figure 5). Although many fibers were intertwined together at the exit side, complete fiber breakage was rarely seen, which could be due to the protection of the outermost UHMWPE layer at the rear face (Appendix A).

2.Failure of PPTA fabrics in PE(8)-KF(9) samples

Figure 7 shows the SEM images captured around a penetration hole on the entrance side (Figure 7a) and the exit side (Figure 7d) of a (u)PE(8)-(u)KF(9) laminate sample. The PPTA fibers showed shear cutting failure on the entrance side around the crater (Figure 7b,c). At the beginning of projectile penetration, a compressive pulse or transient stress wave is induced in the sample ahead of the projectile [49]. When the projectile penetrated the panel and the strain level reached the failure threshold, the local deformation around the penetration hole developed into fiber failure. The PPTA fibers on the exit side of the laminate were pulled out, which is not obvious for the sample (u)PE(9)-(5)KF(8) (Figure 6). The fibers fractured and their ends were split (defibrillate) similar to a brush tip (Figure 7e,f), which differed from the failure mode observed in PE(9)-KF(8) samples (Figure 6). This difference may be due to the absence of protection from the outermost UHMWPE layer on the exit side of the (u)PE(8)-(u)KF(9) sample.

In summary, for both PE(9)-KF(8) and PE(8)-KF(9) samples, the same failure modes of PPTA fibers were identified, i.e., shear cutting failure on the entrance side, and tensile failure on the exit side. However, some differences in detailed morphology were observed. Therefore, placing materials with high compressive and shear strength on the entrance side and materials with high tensile strength on the exit side may improve the ballistic performance of laminate panels.

3.Failure of UHMWPE film

Figure 8 shows SEM images taken around a punched UHMWPE film of a fully penetrated laminate at the entrance side (Figure 8a,b), cross-section (Figure 8c,d), and the exit side (Figure 8e,f) of the sample (u)PE(9)-(5)KF(8). The diameter of the penetration hole can be measured in Figure 8a, which is around 4 mm and much smaller than the diameter of the projectile (5.7 mm), indicating severe friction between the projectile and the sample [23]. Figure 8a–c show the cracks, steps, and shear plugs in the UHMWPE film due to significant compression of its thickness caused by the high impact stress at the entrance side of the laminate. We noticed that there was shear plugging formed in the sample (u)PE(9)-(5)KF(8) during the projectile penetration, as previously reported in the literature [23,51]. The shear plugging is usually generated under the following circumstances: (1) the projectile has sharp edges; (2) the samples exhibit brittle properties; (3) the adhesion between fiber and matrix is high [28]. Based on our experimental results, it is possible that the last two factors contributed to the formation of the shear plugging observed in this study. The UHMWPE region around the crater showed signs of brittle fracture, including sharp-angled edges (Figure 8b), which are very different from typical ductile polymer behavior demonstrating smooth fracture surfaces. The sharp-angled edges of UHMWPE were most likely attributed to the partial crystallinity of UHMWPE, which manifests itself in characteristic splits along the boundaries or easy fracture planes of the crystal structure. On the other hand, as observed in our previous work [42], plasma treatment used in this approach enhanced the adhesion between UHMWPE and PPTA layers, which may also contribute to the formation of shea plugging. Indeed, the sample showed shear plugging behavior (sample (u)PE(9)-(5)KF(8) in Figure 3d) and exhibited the strongest interlayer adhesion between PPTA and UHMWPE among all samples, with an average peeling force of 30.58 N (Figure 3d). The rounded fractured edges of the UHMWPE film (Figure 8c) indicate thermal damage, which was also observed in broken UHMWPE fibers [52]. During the entry of the projectile into the panel, both experienced a sharp increase in temperature due to the resulting friction. Accordingly, an increase in temperature degraded the fracture resistance of UHMWPE. We also noticed a terraced morphology or the striation along the fracture surface of the UHMWPE film (Figure 8b,c), which may be related to the semicrystalline structure of UHMWPE [53,54] and associated with discontinuous crack propagation and accumulated damage [55]. As the bullet continued moving through the laminate structure, the velocity decreased dramatically, the frictional heat reduced, and the temperature dropped below the melting point of the UHMWPE. The UHMWPE film no longer underwent thermal damage and exhibited blunt and clean fracture (yellow arrows in Figure 8d). On the exit side of the sample, the UHMWPE displayed a thick and flat fracture feature due to the tensile stress generated by the large deflection and bending of the laminate panel (Figure 8e,f) [23,28,32,50,56]. The UHMWPE films fail in tension once the dynamic stress exceeded the tensile strength. This result was also observed in UHMWPE fibers [52]. Overall, for UHMWPE films, thermal damage, and brittle failure are the dominant failure mechanisms on the entrance side, and tensile fracture is on the exit side of the composite.

#### 3.2.2. Failure Analysis of a Partially Penetrated Sample

The cross-section of the laminate sample (u)PE(8)-(5)KF(9) around a partially penetrated shot was examined by SEM. The bullet appears to be stopped by the laminate and deformed into a hemispheric shape (Figure 9a). This deformation consumed part of the impact energy. The delamination marked with black arrows in Figure 9a could also have absorbed some impact energy, as discussed in previous sections. It was claimed that delamination usually grows in an area that does not fail under shear or tension and ceases growing once a layer failed [56]. The brighter regions in Figure 9b–f belong to the bullet, which was surrounded by UHMWPE or PPTA layers. In front of the deformed bullet, the PPTA fibers are inlaid in the bullet and pulled by the bullet (Figure 9b,c), which may be caused by copper melting or softening due to friction-induced heat. Figure 9e displays the flat tips of the fractured fibers, suggesting a compressive failure mode where the laminate layers were significantly compressed in front of the bullet [28,49,57]. Figure 9f shows the failed UHMWPE region, the intertwined PPTA fibers, and a large void on the back of the bullet. The light dots present in large quantities on the surface of UHMWPE and PPTA fibers are drops of molten metal and are a result of friction heating of the bullet penetrating through the laminate.

In the open literature, a number of experimental and simulation studies have been carried out to understand the complex process of projectile penetration and energy absorption mechanisms in fabric composites for ballistic applications [20,28,57,58,59,60,61,62,63]. For example, Mawkhlieng et al. [2] classified the energy absorption mechanisms into four categories: fiber and yarn extension, yarn decrimping, fiber and yarn rupture, and yarn pull-out. Naik et al. [61,62] investigated the ballistic impact behavior of two-dimensional woven fabric composites and identified several energy absorption mechanisms, including energy absorbed by the tensile failure of the primary yarns, deformation of the second yarns, matrix cracking, shear plugging, and friction between the projectile and the target. Delamination is another important mechanism for absorbing impact energy, which is well documented in the open literature [21,22,23,61,62] and also confirmed in this study.

Generally, it is believed that there are two stages involved in the projectile penetration process [51]. In the first stage, shear failure occurs due to the high velocity of the projectile at the time of entering the target [49]. During the second stage when the projectile velocity is reduced, tensile failure occurs with bulging or breakout of sub-laminates [22,50]. This general trend has also been observed in the current study, where tensile failure was induced by high tensile stress as a result of large deflection and bending of the laminate panel on the exit side (Figure 6d, Figure 7d and Figure 9a). In the partially penetrated laminates (Figure 9), delamination is clearly observed, as characterized by the triangular void at the right corner of the bullet (Figure 9a).

## 4. Conclusions

In this study, layered PPTA fabric/UHMWPE film laminated samples were fabricated using surface plasma treatment and a hot-pressing process. Their protective performance was evaluated by ballistic tests. The results showed that a moderate increase in adhesion between PPTA fabric and UHMWPE film improved armor performance, which may be attributed to the energy absorption caused by delamination. It has also been found that stacking sequence affected the performance of the composite samples, such that the samples with PPTA fabric placed as the front layer had better protective characteristics. SEM examinations showed that the main failure modes of PPTA fabric were shear cutting failure on the entrance side and tensile failure on the exit side. The failure modes of UHMWPE film were mainly brittle failure and thermal damage at high strain rate under compression on the entrance side, and tensile failure on the exit side of the laminate samples. From the examination of the failure modes, it can be seen that the sample mainly suffered compression and thermal damage on the entrance side and tensile stress on the exit side. Therefore, it is beneficial to the ballistic performance by placing materials with high compressive and shear strength, and high thermal stability on the entrance side, and materials with high tensile strength on the exit side of the composites. In summary, this study reported first-hand in-field bullet testing results of PPTA/UHMWPE composite panels, which can provide important insights for designing, fabricating, and failure analysis of such composite structure for body armors.

## Figures and Tables

**Figure 1 polymers-15-02281-f001:**
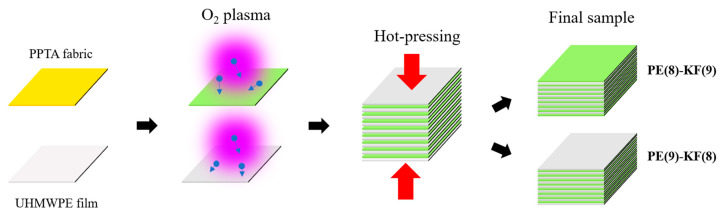
Schematic illustration of the sample fabrication process with two different stacking sequences.

**Figure 2 polymers-15-02281-f002:**
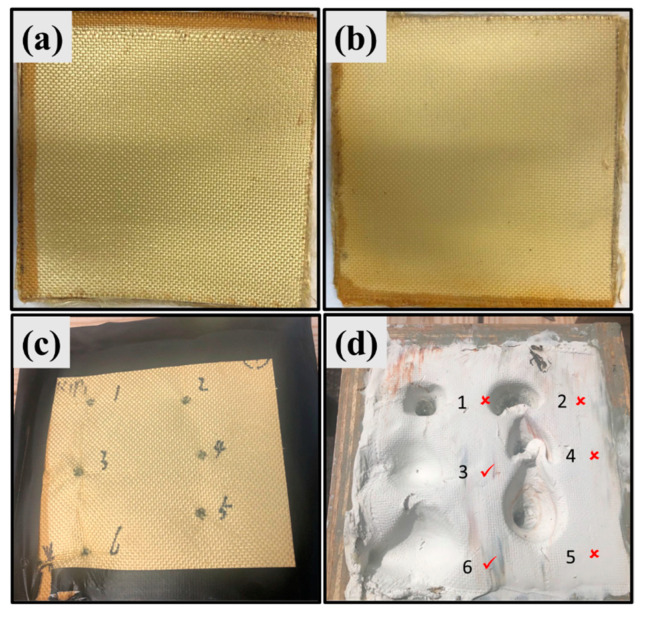
Pictures of UHMWPE/PPTA laminates with stacking sequences of (**a**) eight layers of UHMWPE film and niner layers of PPTA fabric (PE(8)-KF(9)) and (**b**) nine layers of UHMWPE film and eight layers of PPTA fabric (PE(9)-KF(8)); Pictures of a laminate (**c**) and the indents in the backing clay (**d**) after ballistic testing. Numbers 1, 2, 4, 5 indicate completely penetrated shots and numbers 3 and 6 indicate partially penetrated shots.

**Figure 3 polymers-15-02281-f003:**
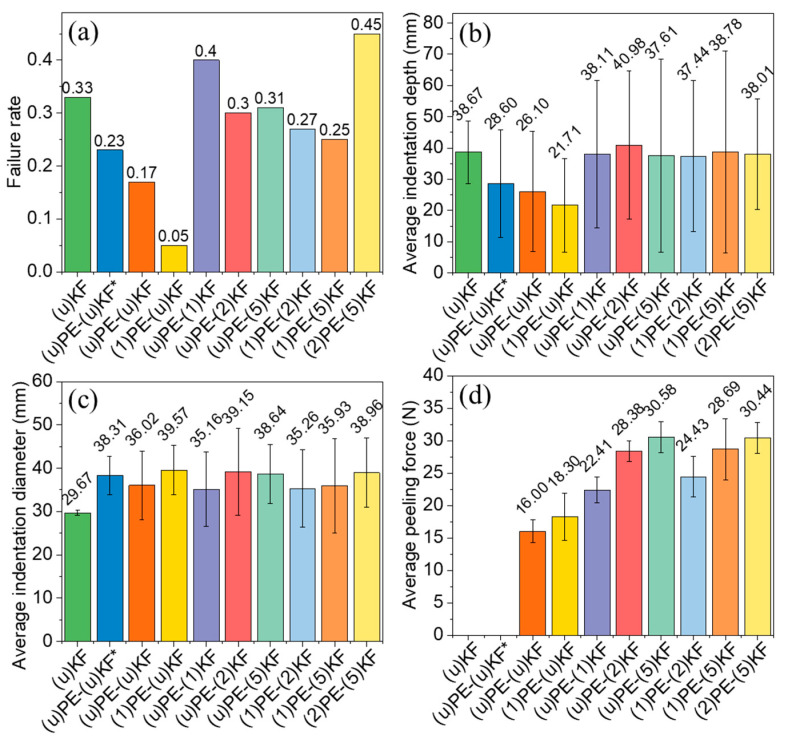
Failure rate (**a**), average indentation depth (**b**), indentation diameter (**c**), and peeling force (**d**) [42] of different samples. (u)KF is the control sample produced by stacking PPTA fabrics without plasma treatment or hot-pressing. (u)PE-(u)KF* is the sample produced by stacking UHMWPE films/PPTA fabrics without plasma treatment or hot-pressing. All other samples were produced by hot pressing eight layers of UHMWPE and nine layers of PPTA fabric. PE: UHMWPE film, KF: PPTA fabric. The numbers in the brackets indicate the plasma treatment time in minutes, and u indicates that the sample was untreated.

**Figure 4 polymers-15-02281-f004:**
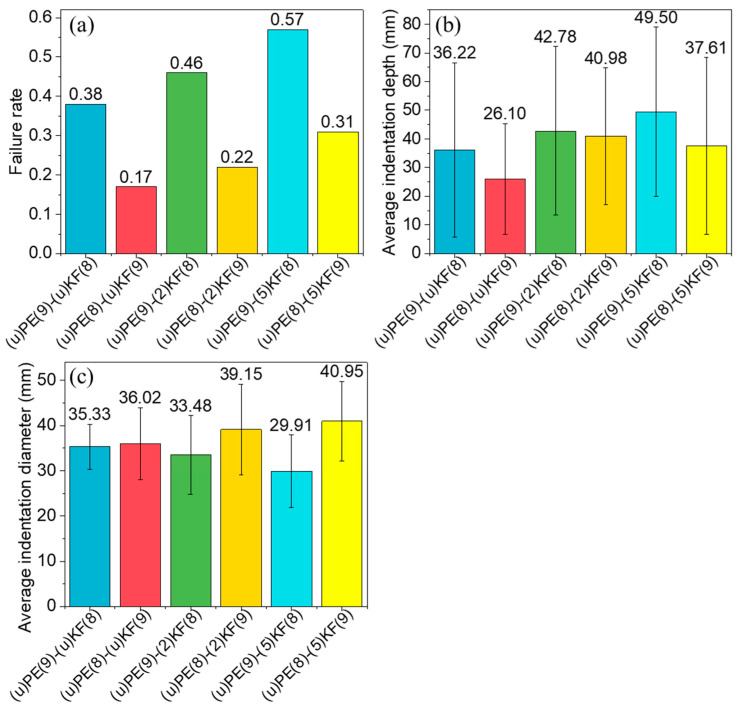
Comparison of the failure rate (**a**), average indentation depth (**b**), and diameter (**c**) between groups of laminate samples with different stacking sequences: PE(9)-KF(8), nine layers of UHMWPE and eight layers of PPTA and PE(8)-KF(9), eight layers of UHMWPE and nine layers of PPTA where PE and KF layers were plasma-treated for a different time (2 or 5 min) or not.

**Figure 5 polymers-15-02281-f005:**
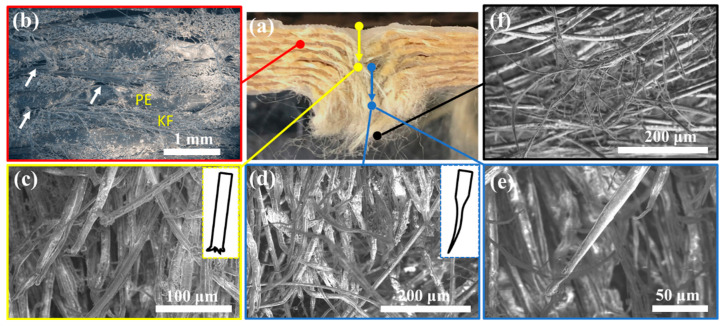
(**a**) A picture showing the cross-section of the panel (u)PE(9)-(u)KF(8) around a completely penetrated shot. (**b**–**f**) SEM images captured at different locations. The white arrows in (**b**) indicate the micro delamination. PE: UHMWPE film, KF: PPTA fabric.

**Figure 6 polymers-15-02281-f006:**
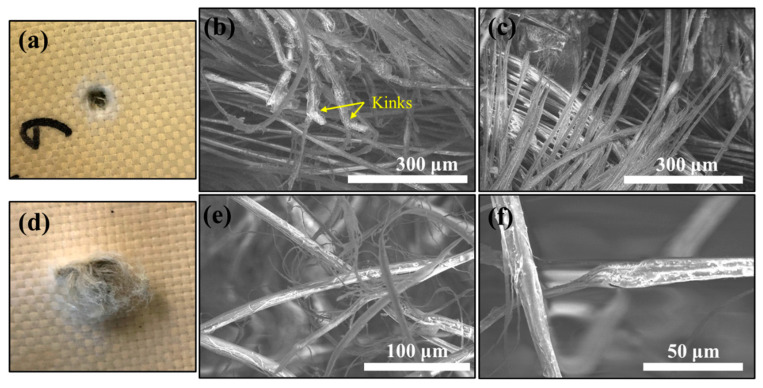
Failure analysis of the laminate sample (u)PE(9)-(5)KF(8) around a completely penetrated hole on the entrance side and exit side. Picture (**a**) and SEM images (**b**,**c**) were captured on the entrance side of the sample. Picture (**d**) and SEM images (**e**,**f**) were captured on the exit side of the sample.

**Figure 7 polymers-15-02281-f007:**
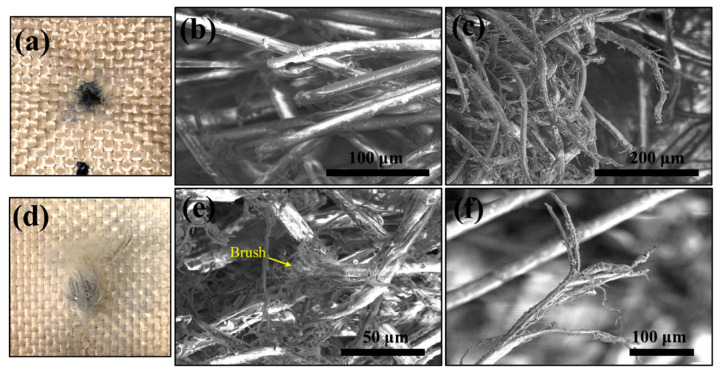
Failure analysis of the laminate sample (u)PE(8)-(u)KF(9) around a completely penetrated hole on the entrance side and exit side of the sample. Picture (**a**) and SEM images (**b**,**c**) were captured on the entrance side of the sample. Picture (**d**) and SEM images (**e**,**f**) were captured on the exit side of the sample.

**Figure 8 polymers-15-02281-f008:**
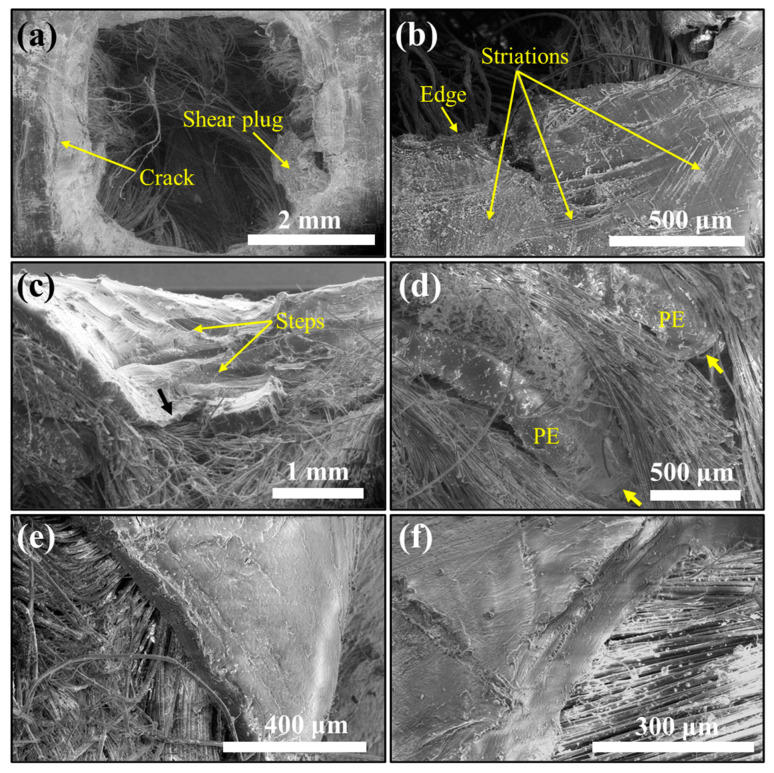
SEM images of the damaged UHMWPE film captured on the entrance side (**a**,**b**), cross-section (**c**,**d**), and the exit side (**e**,**f**) of the laminate sample (u)PE(9)-(5)KF(8) around a hole in a completely penetrated panel.

**Figure 9 polymers-15-02281-f009:**
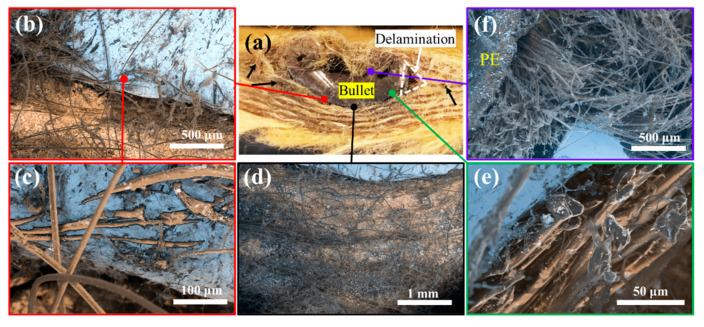
(**a**) A picture showing the cross-section of a laminate sample ((u)PE(8)-(5)KF(9)) around a partially penetrated shot. PE: UHMWPE film, KF: PPTA fabric. (**b**–**f**) SEM images captured at different locations around the bullet.

## Data Availability

The data presented in this study are available on request from the corresponding author.

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
