# Peer review of "Anti-Ballistic Performance of PPTA/UHMWPE Laminates"

_polymers, 2023, doi:10.3390/polym15102281_

Round 1
Reviewer 1 Report
This study examined the fabrication and ballistic performance of layered composite materials, consisting of Poly(p-phenylene terephthalamide) (PPTA) fabrics and ultra-high-molecular-weight polyethylene (UHMWPE) films, using plasma treatment and hot-pressing.
The anti-ballistic performance of the composite was assessed based on failure rate, penetration depth, indentation diameter, and peeling force, while the failure mode was characterized through photographs and SEM images.
Although the topic is interesting, the current manuscript lacks an in-depth analysis of the reinforcing mechanism or material properties, despite evaluating interlayer adhesion in a previous study. To meet the standards of a journal publication, it is recommended that the authors conduct additional experiments on material properties or characterizations.
The significant standard deviation for the indentation depth warrants further investigation. Can you provide an explanation or conduct additional tests to reduce this variation?
Author Response
Reviewer 1:
Comments and Suggestions for Authors
This study examined the fabrication and ballistic performance of layered composite materials, consisting of Poly(p-phenylene terephthalamide) (PPTA) fabrics and ultra-high-molecular-weight polyethylene (UHMWPE) films, using plasma treatment and hot-pressing.
The anti-ballistic performance of the composite was assessed based on failure rate, penetration depth, indentation diameter, and peeling force, while the failure mode was characterized through photographs and SEM images.
Comment: Although the topic is interesting, the current manuscript lacks an in-depth analysis of the reinforcing mechanism or material properties, despite evaluating interlayer adhesion in a previous study. To meet the standards of a journal publication, it is recommended that the authors conduct additional experiments on material properties or characterizations.
Response: We thank the reviewer for the comments. As the reviewer suggested, we included additional discussion on the reinforcement mechanisms to enhance the scientific aspect of this work (the last paragraph of the Introduction, the last paragraph of Sample preparation, and the second paragraph of 3.1.1. Effect of interlayer adhesion section). On the other hand, due to the long sample preparation and testing involved in this type of sample, it is difficult to generate additional meaning data within the short response time. We plan to conduct more experiments including material characterizations in the future work.
Comment: The significant standard deviation for the indentation depth warrants further investigation. Can you provide an explanation or conduct additional tests to reduce this variation?
Response: The measurement results for indentation depth and diameter did show large variation, likely due to the nature of the bullet testing that was conducted in the field. An automated test in a well-controlled environment would be highly desired. But the research team doesn’t have access to such facilities. On the other hand, conducting additional large number of tests could reduce the standard deviation. However, the sample fabrication and testing cycle are very long (~ one month), cannot be conducted in a short period of time.
Reviewer 2 Report
Although the paper is of interest, it would benefit if the authors could comment on the findings of the most relevant papers quoted. More recent references would be welcome.
Please make the following edting:
1 - Page 4: where is "...is another vital indictor to assess..." write "...is another vital indicator to assess..."
2 - Page 10: where is "... whichwas also observed..." write "... which was also observed..."
3 - Page10: whre is "...assoiciated with discontinuous crack..." write "...associated with discontinuous crack..."
Author Response
Reviewer 2:
Comments and Suggestions for Authors
Comment: Although the paper is of interest, it would benefit if the authors could comment on the findings of the most relevant papers quoted. More recent references would be welcome.
Response: We thank the reviewer for the suggestions. We added more discussion and references on more recent findings in the introduction section. For example: in the Introduction section, we added “Although significant progress has been made in improving the performance of soft body armor by adopting high-performance fabrics, there is still a great challenge in developing body armor with low bulge deformation, antiballistic capabilities, and flexibility at the same time [3,4]. In recent years, nanomaterials, such as carbon nanotubes (CNTs) and graphene, which are among the stiffest and strongest materials, have been explored as reinforcements for armor composites [5,6]. New designs and material systems are highly desired for further improvement [7].”
Comment: Please make the following edting:
1 - Page 4: where is "...is another vital indictor to assess..." write "...is another vital indicator to assess..."
2 - Page 10: where is "... whichwas also observed..." write "... which was also observed..."
3 - Page10: whre is "...assoiciated with discontinuous crack..." write "...associated with discontinuous crack..."
Response: The suggested changes have been made in the revised manuscript.
Reviewer 3 Report
Dear Authors,
The manuscript entitled "Anti-ballistic performance of PPTA/UHMWPE laminates" submitted for review by Long Zhu, Weixiao Gao, Dmitriy A. Dikin, Simona Percec, Fei Ren from Temple University in Philadelphia concerns ballistic tests layered PPTA fabric/UHMWPE film laminated samples obtained using surface plasma treatment and hot-pressing process. In general, the manuscript is interesting, has a correct structure and a high scientific level.
1. The novelty of the work is properly shown, but the authors have lost the formulation of the research problem and the purpose of the work, as well as the conclusions regarding the contribution of their work to the development of the discipline/scientific field they represent.
2. Also note the consistent use of SI units throughout the manuscript (inches and feet should be converted).
3. It is also advisable to include a paragraph in the introduction on "smart polymers", the reaction of which depends on the action, and from which such products are already made in the form of lightweight protective body armor.
After taking these corrections into account, the manuscript can be published in the journal: Polymers.
Best regards,
Author Response
Reviewer 3:
Comments and Suggestions for Authors
Dear Authors,
The manuscript entitled "Anti-ballistic performance of PPTA/UHMWPE laminates" submitted for review by Long Zhu, Weixiao Gao, Dmitriy A. Dikin, Simona Percec, Fei Ren from Temple University in Philadelphia concerns ballistic tests layered PPTA fabric/UHMWPE film laminated samples obtained using surface plasma treatment and hot-pressing process. In general, the manuscript is interesting, has a correct structure and a high scientific level.
Comment: 1. The novelty of the work is properly shown, but the authors have lost the formulation of the research problem and the purpose of the work, as well as the conclusions regarding the contribution of their work to the development of the discipline/scientific field they represent.
Response: We thank the reviewer for the comments. We added more discussion at the end of the Introduction to illustrate the purpose of this work. We also added more information at the end of the Abstract and at the end of the Conclusion section to highlight this work’s contribution.
Comment: 2. Also note the consistent use of SI units throughout the manuscript (inches and feet should be converted).
Response: We have converted all units to SI units.
Comment 3. It is also advisable to include a paragraph in the introduction on "smart polymers", the reaction of which depends on the action, and from which such products are already made in the form of lightweight protective body armor.
Response: We thank the reviewer for the suggestions. We added one paragraph in the introduction section (second paragraph in the Introduction section) to discuss application of “smart polymers” or smart materials on multifunctional soft body armor, especially shear-thickening gels, whose stiffness increases dramatically at high stain rate.
Comment: After taking these corrections into account, the manuscript can be published in the journal: Polymers.
Reviewer 4 Report
Fei Ren et al. in their manuscript entitled “Anti-ballistic performance of PPTA/UHMWPE laminates”, prepared poly(p-phenylene terephthalamide) (PPTA) and ultra-high-molecular-weight polyethylene (UHMWPE) composites for body armor applications using plasma treatment and hot-pressing processes. Based on provided ballistic tests, they reported that the ballistic performance of their prepared composites was improved. However, some deficiencies listed below are observed:
1. The manuscript contains some typos/spelling mistakes. They should be corrected.
2. The novelty of the work is not clear. Why do we need to layered composites from PPTA fabrics and UHMWPE films as a new design for body armor applications?
3. The main lack of the manuscript is the absence of the investigation of how the applied plasma treatment and hot-pressing processes will affect the structure and mechanical properties for each layer (PPTA and UHMWPE). The structure morphology and all mechanical and physical properties will be changed after applying these processes. SEM, FT-IR, mechanical properties, etc. for UHMWPE and PPTA after plasma treatment and hot-pressing processes are absent. However, the authors carried out some investigations related to their composites in their previous article [https://doi.org/10.3390/polym13162600], but they should connect this manuscript with the previous results.
4. The mechanical tests for investigating the adhesion strength between the PPTA and UHMWPE are absent.
5. Based on Figure 4(b), the average indentation depth for all samples is almost the same (all values are in the range of the standard deviation). Can the authors explain this result?
6. The authors examined two factors affecting the impact performance of soft body armors, which were the interlaminar adhesion and stacking sequence. However, the obtained results in this section cannot be surprisingly accepted while there are also other important parameters that should be addressed.
7. The authors wrote “The shear plugging is usually generated under the following circumstances: (1) the projectile has sharp edges; (2) the samples exhibit brittle properties; (3) the adhesion between fiber and matrix is high [18]. The UHMWPE region around the crater showed signs of brittle fracture, including sharp-angled edges (Figure 8b), which are very different from typical ductile polymer behavior demonstrating smooth fracture surfaces. We attribute these features to the partial crystallinity of UHMWPE which manifests itself in characteristic splits along the boundaries or easy fracture planes of the crystal structure”. How can it be related to the semicrystalline structure of UHMWPE? Moreover, there are no clear results or investigations about the adhesion between fiber and matrix. How did the authors find that this adhesion was high? The discussion here is not acceptable.
In general, the article is weak, and it is more technical than scientific. The authors should improve the scientific level of the manuscript.
Author Response
Reviewer 4:
Comments and Suggestions for Authors
Fei Ren et al. in their manuscript entitled “Anti-ballistic performance of PPTA/UHMWPE laminates”, prepared poly(p-phenylene terephthalamide) (PPTA) and ultra-high-molecular-weight polyethylene (UHMWPE) composites for body armor applications using plasma treatment and hot-pressing processes. Based on provided ballistic tests, they reported that the ballistic performance of their prepared composites was improved. However, some deficiencies listed below are observed:
Comment 1. The manuscript contains some typos/spelling mistakes. They should be corrected.
Response: We carefully edited the manuscript and tried our best to eliminate all typos and spelling errors.
Comment 2. The novelty of the work is not clear. Why do we need to layered composites from PPTA fabrics and UHMWPE films as a new design for body armor applications?
Response: We thank the reviewer for the comments. In this study, we prepared layered composites from PPTA fabrics and UHMWPE films with controlled interlay adhesion by plasma treatment and hot pressing. We believe the different interlay adhesion can induce different levels of delamination process, which absorb impact energy when the composite was subjected to high-speed projectile, thus improving the anti-ballistic performance.
Comment 3. The main lack of the manuscript is the absence of the investigation of how the applied plasma treatment and hot-pressing processes will affect the structure and mechanical properties for each layer (PPTA and UHMWPE). The structure morphology and all mechanical and physical properties will be changed after applying these processes. SEM, FT-IR, mechanical properties, etc. for UHMWPE and PPTA after plasma treatment and hot-pressing processes are absent. However, the authors carried out some investigations related to their composites in their previous article [https://doi.org/10.3390/polym13162600], but they should connect this manuscript with the previous results.
Response: We agree with the reviewer that the characterization of individual layers after plasma treatment and hot pressing is important. As the reviewer pointed out, we performed certain characterization in our previous work [32]. To reflect the connection between current and the previous works, we added more description in section 2.2.
Comment 4. The mechanical tests for investigating the adhesion strength between the PPTA and UHMWPE are absent.
Response: We thank the reviewer for the comments. When the composites were not subjected to hot pressing process, there was no interlayer adhesion. In Figure 3(d), samples (u)KF and (u)PE-(u)KF* were produced by simply stacking PPTA fabrics and/or UHMWPE films without hot-pressing. Therefore, there is no interlayer adhesion strength (average peeling force) between the layers.
Comment 5. Based on Figure 4(b), the average indentation depth for all samples is almost the same (all values are in the range of the standard deviation). Can the authors explain this result?
Response: Yes, the measurement results for indentation depth and diameter did show very large variation. As an example, from the indent of unpenetrated shots #3 and 6 on the backing clay (Figure 2(d)), although both are unpenetrated shot, the indentation diameters showed a big difference. These results could be confirmed by repeating more samples and reducing the number of shots on each sample. However, due to the in-field nature of the test, we think testing on more samples may provide marginal benefit to reduce the variation. We think the most important parameter for evaluating the anti-ballistic performance is failure rate in this study, and the indentation depth and indentation diameter were recorded as supplementary data for evaluating the anti-ballistic performance.
Comment 6. The authors examined two factors affecting the impact performance of soft body armors, which were the interlaminar adhesion and stacking sequence. However, the obtained results in this section cannot be surprisingly accepted while there are also other important parameters that should be addressed.
Response: We agree that there are many other parameters affecting the ballistic performance, including (a) material parameters, such as fiber modulus, tenacity, density, and yarn to yarn friction; (b) structural parameters, such as number of layers, yarn twist, thread density, weave; (c) projectile parameters, such as mass, shape and velocity, and (d) testing parameters, such as shooting location, angle boundary conditions, and number of shots, etc. However, this study was limited by two major parameters – time constraints and funding. Both PPTA and UHMWPE are expensive materials, and access to shooting tests is limited. Although we believe that the results of field tests already deserve to be published, we continue our experimental work to explore a larger set of parameters.
Comment 7. The authors wrote “The shear plugging is usually generated under the following circumstances: (1) the projectile has sharp edges; (2) the samples exhibit brittle properties; (3) the adhesion between fiber and matrix is high [18]. The UHMWPE region around the crater showed signs of brittle fracture, including sharp-angled edges (Figure 8b), which are very different from typical ductile polymer behavior demonstrating smooth fracture surfaces. We attribute these features to the partial crystallinity of UHMWPE which manifests itself in characteristic splits along the boundaries or easy fracture planes of the crystal structure”. How can it be related to the semicrystalline structure of UHMWPE? Moreover, there are no clear results or investigations about the adhesion between fiber and matrix. How did the authors find that this adhesion was high? The discussion here is not acceptable.
Response: We agree with the reviewer that this discussion was not well presented. The sharp-angled edges of UHMWPE were probably attributed to the partial crystallinity of UHMWPE which manifests itself in characteristic splits along the boundaries or easy fracture planes of the crystal structure. We revised this section as follows:
“The shear plugging is usually generated under the following circumstances: (1) the projectile has sharp edges; (2) the samples exhibit brittle properties; (3) the adhesion between fiber and matrix is high [18]. Based on our experimental results, it is possible that the last two factors contributed to the formation of the shear plugging observed in this study. The UHMWPE region around the crater showed signs of brittle fracture, including sharp-angled edges (Figure 8b), which are very different from typical ductile polymer behavior demonstrating smooth fracture surfaces. surfaces. The sharp-angled edges of UHMWPE were probably attributed to the partial crystallinity of UHMWPE which manifests itself in characteristic splits along the boundaries or easy fracture planes of the crystal structure. On the other hand, as observed in our previous work [32], plasma treatment used in this approach had enhanced the adhesion between UHMWPE and PPTA layers, which may also contribute to the formation of shea plugging. Indeed, the sample showed shear plugging behavior (sample (u)PE(9)-(5)KF(8) in Figure 3(d)) exhibited the strongest interlayer adhesion between PPTA and UHMWPE among all samples, with an average peeling force of 30.58 N (Figure 3(d)).”
Comment: In general, the article is weak, and it is more technical than scientific. The authors should improve the scientific level of the manuscript.
Response: We appreciate the overarching and detailed comments of the reviewers. After revising the manuscript based on the reviewers’ comments, we strengthened the discussions relating material performance to the properties, structures, and processing at various places in the manuscript. We believe that the manuscript has been significantly improved and should be of interest to both engineers and scientists working in the field of composites.
Round 2
Reviewer 1 Report
Thank you for the proper revision. I suggest this manuscript can be accepted in its present form.
